# Physical Properties and Storage Stability of Buton Rock Asphalt Modified Asphalt

**DOI:** 10.3390/ma15103592

**Published:** 2022-05-18

**Authors:** Yue Su, Xiaodi Hu, Jiuming Wan, Shaopeng Wu, Yinglong Zhang, Xing Huang, Zhangjun Liu

**Affiliations:** 1School of Civil Engineering and Architecture, Wuhan Institute of Technology, Wuhan 430205, China; 13235670277@163.com (Y.S.); huxiaodi@wit.edu.cn (X.H.); zhangyl@stu.wit.edu.cn (Y.Z.); liuzhangjun@wit.edu.cn (Z.L.); 2State Key Laboratory of Silicate Materials for Architecture, Wuhan University of Technology, Wuhan 430070, China; wusp@whut.edu.cn; 3Poly Changda Engineering Co., Ltd., Guangzhou 510062, China; huangx202205@163.com

**Keywords:** Buton rock asphalt, storage stability, particle size, physical property

## Abstract

Buton Rock Asphalt (BRA) refers to the natural rock asphalt natively produced on the Buton island of Indonesia. It is often used as a modifier to enhance the performance of asphaltpavement. However, the segregation of BRA in BRA-Modified Asphalt (BRA-MA) has restricted its application. This study aims to investigate how the particle size and content of BRA affect the physical properties and storage stability of BRA-MA. Penetration, softening point, viscosity, and viscosity-temperature susceptibility (VTS) were analyzed. The evaluation method of storage stability was discussed and determined. The segregation of BRA in BRA-MA of static storage and transportation process were simulated and tested. The results suggest that the softening point and viscosity were positively correlated to BRA content and inversely determined by particle size. Penetration, VTS, and ductility were reduced due to the decline in particle size and increment of BRA content. The index of segregation value based on viscosity difference showed better statistical and quantitative significances than the softening-point difference in evaluating the storage stability. The particle size and content of BRA are positively correlated to the segregation of BRA-MA. Both the storage temperature and time were positively correlated to the segregation of BRA-MA. We prove that the relationship between specific surface area and segregation are power functional. BRA-MA with BRA whose 50% particle sizes are lower than 13.6 μm showed low segregation in transportation.

## 1. Introduction

Natural rock asphalt refers to the asphalt that exists in rock crevices, which is often used as an additive and modifier for asphalt materials [1,2,3,4,5]. Buton Rock Asphalt (BRA) is a kind of rock asphalt that originates from Buton Island, Sulawesi province, Indonesia [6,7]. It is used as a modifier to prepare Buton Rock Asphalt-Modified Asphalt (BRA-MA) to improve high-temperature and moisture performances of pavement [6,7]. The preparation methods of BRA-modified asphalt mainly include wet processing and dry processing. Wet processing uses BRA as a modifier to prepare BRA-modified asphalt, whose performance is significantly improved compared to base asphalt [8]. Lv et al. [7] reported that the addition of BRA could help to improve the aging resistance of bio-asphalt. The use of BRA can reduce the consumption of petroleum asphalt and cost of materials. It is suggested that BRA is also helpful for the cleaner and cheaper production of asphalt pavement construction, signifying that the application of BRA is beneficial for environment-protection [9] and economic values [10,11,12,13]. Consequently, the application prospect of BRA-MA is very extensive.

However, the storage stability of BRA-MA requires further study since the segregation of BRA often occurs in the production plant and transportation of BRA-MA, especially in wet processing. Hence, the storage stability of BRA-MA can be divided into two parts: static storage [14] in the production plant and the transportation-storage process [15]. Segregation characteristic of the two parts is differential. Thus, the storage stability of BRA-MA during thermal static storage and transportation should be investigated and improved to address the segregation of BRA. Direct observation method, softening-point difference [16], Fourier infrared spectroscopy, and fluorescence microscope were used to evaluate the storage stability of modified asphalt [17]. Gu et al. [18] evaluated the storage stability of waterborne epoxy emulsified asphalt by fluorescence microscope. It was found that the content increase in waterborne-epoxy resin improves the compatibility of waterborne epoxy resin emulsified asphalt. Nevertheless, whether the storage stability evaluation method of polymer-modified asphalt is applicable to BRA-MA is not clear, since BRA is different with polymer modifier. For instance, it is quite difficult to be distinguished under a microscope, thus the micro-observation method is not applicable.

The factors [19] affecting the storage stability of modified asphalt include the content of modifier, storage temperature, storage time, and particle size. T J, Navarro et al. [20] and Liang et al. [21] studied the effects of the storage temperature and sample particle size on the storage stability of tire-rubber powder modified asphalt. It was found that the storage stability of tire-rubber-powder-modified asphalt gradually decreases with the increase in storage temperature and rubber powder particle size. Fang et al. [22] studied the effect of the preparation process on the storage stability of waste polyethylene-modified asphalt by using differential scanning calibration (DSC) and thermal gravimetric analysis (TGA). The preparation process was found to have little effect on the storage stability of modified asphalt. However, how particle size and BRA content determine the storage stability of BRA-MA has not been systematically studied. The effect of storage temperature, storage time, and cooling process on static storage and transportation storage is not yet explicit.

BRA powders of five different sizes were used as modifiers, and BRA-MA with 10%, 20%, 30%, and 40% BRA are introduced in this study. Penetration, softening point, ductility, viscosity, and viscosity-temperature susceptibility are characterized to understand the effect of particle size and BRA content on their physical properties. The softening-point difference test and viscosity difference test are separately conducted and discussed. Therefore, the appropriate evaluation method for the segregation of BRA-MA can be determined. Static storage stabilities by different temperature and time dependencies are characterized to illustrate the static storage process in the production plant before transportation. Storage stability during transportation is also tested in a simulation of the cooling process in a container in the vehicle. It provides a scientific basis for improving the physical properties and storage stability of BRA-MA.

## 2. Materials and Experimental Methods

### 2.1. Materials

Part a of Figure 1 illustrates the appearance of BRA, which is brown powder after being crushed. Figure 1b presents the mesoscopic image of BRA through Scanning Electron Microscope analysis, whose magnification is 4500. The image suggests that BRA has a very rough surface and porous structure. Rock asphalt and mineral cannot be clearly distinguished through the mesoscopic image; it proved that rock asphalt was evenly mixed and combined with minerals. The asphalt content of BRA was tested based on the ignition oven method [23], since BRA can be regarded as a mixture of asphalt and mineral. The result shows that the mass percentage of asphalt in BRA is 27.8%, while the minerals account for 71.2%. The moisture content of BRA is 1%. This study also introduced base asphalt of Pen 60~80, which was produced by the Guochuang company, Hubei province, as the material for preparing the BRA-MA. Table 1 illustrates the properties of base asphalt. Penetration, softening point, ductility, and viscosity are characterized according to JTG E20-2011 and JTG F40-2004. 

### 2.2. Preparation of BRA-MA

BRA should be pre-treated before preparing various BRA-MA. BRA raw material should be kept in a temperature-controlling box at 80 °C for 12 h. Moisture in BRA can therefore be removed. BRA was then ground into five different particle sizes using a planetary ball mill. A Malvern laser particle-size analyzer [21] was used to characterize the particle-size distribution of the five BRA samples. The five BRA samples were named as BRA-1, BRA-2, BRA-3, BRA-4, and BRA-5, respectively. Table 2 presents the particle-size testing results of BRA samples; corresponding d(0.5) particle sizes are 6.26 μm, 9.55 μm, 12.58 μm, 13.60 μm, and 106.22 μm, respectively. The d(0.5) refers to the volume content of particles accounts for 50% whose particle size is smaller than a particle size. It is used to reflects the particle size. Specific surface area was also presented, which were negatively correlated to the particle size. 

Base asphalt should be heated to 155 °C in an agitator for preparing BRA-MA. Rotational speed was set as 2500 r/min for 20 min after half of the BRA was added to the base asphalt. Then, the remaining BRA was added to the base asphalt, while the speed of a high-speed shear agitator was also set as 2500 r/min for another 20 min. Figure 2 shows the high-speed shear agitator for BRA-MA preparation. This special mixing method can ensure the uniformity of BRA in BRA-MA, while the particle size of BRA remained unchanged during shearing.

### 2.3. Experimental Methods

This study introduced the external-adding method, namely, the mixing proportion is the mass ratio of BRA to base asphalt. The proportions of BRA 1–5 in corresponding BRA-MA were 10%, 20%, 30%, and 40%, respectively. Thus, 20 BRA-MA samples were prepared based on different particle sizes and contents. Figure 3 illustrates the outline of this study. BRA-MA’s physical properties, storage-stability evaluation method, static storage stability, and storage stability of transportation process were studied.

Firstly, penetration, softening point, ductility, and viscosity of base asphalt and BRA-MA were characterized. The impact of BRA content and particle size on physical properties can be concluded. Secondly, this study introduced a separating-tube method to indicate the distribution uniformity of BRA in BRA-MA. Two indicators for evaluating storage stability were compared, so that more appropriate evaluation indicators can be obtained. Subsequently, the stability of BRA-MA in factory static storage were investigated. Finally, the storage stability of BRA-MA during the transportation process was simulated according to temperature decline in transportation.

The storage stability of modified asphalt is the key technical requirement for its production and transportation, which may lead to modification failure and certainly affect the service performance of the modified asphalt mixture. In order to study the influence of particle size and BRA content on the storage stability of BRA-MA, the optimum indicator for storage-stability evaluation should be determined. As explained in stage 5 of the outline of this study in Figure 3, the softening-point difference and index of segregation (IS) based on the viscosity difference between the top and bottom part samples were used to characterize the segregation degree of BRA. The two evaluation methods were used separately to analyze the influence of particle size and BRA content on the storage stability of BRA-MA. The feasibility of the two indicators of the methods was discussed. The static significance of the results according to two indicators was analyzed by variance tests. If F is greater than F crit, then the difference between the results is proven. If the *p*-value is greater than 0.01 and less than 0.05, the difference is significant. If the *p*-value is less than 0.01, the difference is highly significant.

Figure 4 implies the storage process of BRA-MA in a production plant and during transportation. The produced BRA-MA was contained in a soaking tank with temperature-controlling equipment, which can keep BRA-MA at a relatively stable high temperature. Static storage at a high temperature of BRA-MA can save the time and cost of reheating, especially in short-term construction. Maintaining the high liquidity of BRA-MA facilitates rapid loading onto a transport vehicle. Thus, the storage of BRA-MA in a production plant is static. The static storage of BRA-MA refers to the hot-storage process [22,24,25] in a production plant before transportation. Then, BRA-MA is pumped into the vehicle container for transportation, the transportation time is usually less than 12 h. Both the storage stability of BRA-MA in the production plant and transportation process storage was discussed in this study.

The separating tube used in this study was composed of an aluminum sheet with a thickness of 0.4 mm. Therefore, the customized tube was easy to cut and separate. It should be averagely separated and noted as 3 parts by length, which were the top, middle, and bottom parts. Figure 5 expresses how the separating-tube method was conducted to evaluate the storage stability and determination of the evaluation indicator for segregation. At stage 1, the separating tube should be filled with BRA-MA on a specific shelf, exactly when the mixing of BRA and base asphalt was completed. Subsequently, the separating tubes should be kept vertically in a temperature-controlling box according to the established storage conditions, during which the segregation occurs. BRA would gradually settle at the bottom of the tubes due to its gravity. In the next stage, segregated samples were kept in a freezer at −4 °C in 4 h. The segregation rate of BRA would decrease to a very low level due to the low temperature, which can be considered that the segregation of BRA-MA was terminated. As shown in stage 4, separating tubes should be cut and separated into 3 parts by an electric saw. Subsequently, BRA-MA from the top and bottom parts should be characterized by the softening-point and viscosity tests, respectively. Softening point and viscosity of the top and bottom samples were different, as a result of the BRA content difference of the two samples owing to BRA segregation. In the final stage, the softening-point difference and index of segregation based on viscosity difference can be acknowledged, respectively. The optimum evaluation indicator can be therefore be determined. The storage stability of BRA-MA can consequently be calculated and obtained.

## 3. Results and Discussions

### 3.1. Physical Properties of BRA-MA

#### 3.1.1. Softening Point

Figure 6 presents the results of the softening point of base asphalt and BRA-MA. BRA-MA samples were labeled with BRA type and BRA content. The yellow bars present the data of base asphalt, whose BRA content is 0%. The softening point of BRA-MA showed an obvious rising tendency, along with the increase in BRA content. The softening-point value of BRA-MA with 40% BRA-1 was 54.05 °C, which was 5.3 °C higher than that of base asphalt. On the other hand, the softening point of BRA-MA with the same BRA content decreased from BRA-1 to BRA-5. The results of BRA-MA can meet the standard limits of JTG F40-2004. It proved that the softening point of BRA-MA can be enhanced by a decline in BRA particle size. The results also illustrate that the softening point is positively correlated to BRA content. The high-temperature stability and temperature sensitivity of asphalt is generally evaluated by its softening point [26,27]. The temperature sensitivity of asphalt can be reduced by the addition of BRA. Thus, the increase in BRA content and decline in particle size would help to enhance the high-temperature performance of BRA-MA. 

#### 3.1.2. Penetration

Figure 7 shows the penetration of base asphalt and BRA-MA. Penetration indicates the hardness of asphalt. A higher penetration value means a lower hardness. The yellow bars present the data of base asphalt without BRA. Penetration of BRA-MA decreased as BRA content increased, while base asphalt presented the highest penetration value. Penetration of BRA-MA containing 40% BRA-1 was 4.85 mm, which decreased by 31.9% compared to that of base asphalt. Additionally, the penetration of BRA-MA showed an approximativelyincreasing trend from BRA-1 to BRA-5 at the same BRA content. Therefore, the penetration of BRA-MA increased with the increment in particle size. The results suggest that BRA content negatively affects penetration, while the particle size of BRA shows a contrary effect. Therefore, the addition of BRA can improve the hardness of asphalt.

#### 3.1.3. Ductility

Figure 8 presents the 15 °C ductility of base asphalt and BRA-MA, whose yellow bars are the data of base asphalt. It is obvious that the ductility of base asphalt is about 150 cm, while ductility values of BRA-MA are almost lower than 60 cm. The ductility value of BRA-MA with 40% BRA-5 was 18.5 cm, whose decline rate reached 84.6% compared to base asphalt. This result indicates that the addition of BRA leads to remarkable ductility loss. The ductility of BRA-MA also shows a downward trend as the increment in BRA content and particle size. Thus, BRA content and particle size negatively determine the ductility of BRA-MA. It is believed that it was the inorganic mineral particles in BRA that played a critical role in reducing the ductility of BRA-MA. Although its ductility is much lower than that of base asphalt, whether its low-temperature performance is poor is still hard to define due to the lack of corresponding standard for BRA.

#### 3.1.4. Viscosity and Viscosity-Temperature Susceptibility (VTS)

Viscosities of 20 BRA-MA at 115 °C, 135 °C, 155 °C, and 175 °C were measured by a Brookfield viscometer. Figure 9, Figure 10, Figure 11 and Figure 12 illustrate the viscosity of BRA-MA by temperature and particle-size dependency. The viscosity of base asphalt was appended for comparison. The 135 °C viscosity results of BRA-MA can meet the JTG F40-2004 standard limits for modified asphalt. The viscosity of base asphalt was significantly lower than that of BRA-MA. BRA-MA with BRA-1 showed the highest viscosity at the same temperature and BRA content. BRA-MA with 40% BRA-1 showed the highest viscosity, and its growth rate reached 114% compared to that of base asphalt. The viscosity difference between base asphalt and BRA-MA gradually decreased with the increase in temperature. The larger particle size of BRA led to lower viscosity. However, the effect of BRA particle size of viscosity was not very significant when the temperature was over 135 °C. Therefore, the viscosity of asphalt apparently increases with the addition of BRA and a smaller particle size. 

Figure 13 explains the effect of BRA content on the viscosity of BRA-MA, which uses a logarithmic longitudinal axis. The viscosity result of BRA-MA with 40% BRA-1 was used for an instance, and corresponding fitting curves of the data were presented. As the BRA-1 content increased from 10% to 40%, the viscosity of BRA-MA showed an obvious rising trend. This result proves that raising BRA content contributes to promote the viscosity of BRA-MA. To conclude, viscosity had a negative correlation with the particle size of BRA, but it was positively affected by BRA content. 

VTS [28] refers to the slope of the viscosity–temperature curve with the iterated logarithm of viscosity and logarithm of temperature as the coordinate axes [29]. For instance, Figure 14 demonstrates the straight lines of BRA-MA fitting with BRA-1. According to ASTM D 2493, the smaller absolute value of VTS implies the lower temperature sensitivity of asphalt [30]. The basic equation of VTS is:(1)$$\mathrm{VTS}=\frac{\mathrm{lg}\left[\mathrm{lg}\left(\eta_{T_{2}}\right)\right]-\mathrm{lg}\left[\mathrm{lg}\left(\eta_{T_{1}}\right)\right]}{\mathrm{lg}\left(T_{2}\right)-\mathrm{lg}\left(T_{1}\right)}$$
where the T_1_ and T_2_ = temperatures (centigrade) of the binder at two known points, and $\eta_{T_{1}}$ and $\eta_{T_{2}}$ 5 = viscosities of the binder at the same two points. The viscosities of BRA-MA at 115 °C, 135 °C, 155 °C, and 175 °C were used to fit a straight line according to Equation (1).

The absolute value of BRA-MA VTS values can be observed in Table 3, which involves the VTS value of base asphalt for comparison. The addition of BRA can reduce the VTS of asphalt. It is clear that the absolute VTS value decreased with the increase in BRA content, which indicates that BRA can reduce the temperature susceptibility of asphalt. Additionally, the smaller particle size also leads to a lower absolute VTS value. Therefore, VTS is negatively correlated to BRA content, but positively determined by particle size. These results indicate that the higher BRA content and smaller particle size reduce BRA-MA’s temperature susceptibility, which may improve the BRA-MA mixture’s stability at high temperatures.

### 3.2. Storage-Stability Evaluation Determination

#### 3.2.1. Softening Point Difference

Softening-point difference [31] is one of the most commonly used indicators in the storage-stability evaluation of polymer-modified asphalt. Figure 6 proves that BRA content is positively correlated to the softening point of BRA-MA; thus, a higher softening point means a higher BRA content. Through the difference of the softening point between the top and bottom samples, the segregation of BRA content between the top and bottom of the separating tube can be determined. If the softening-point difference between the top and bottom is less than 2.2 °C, according to ASTM D 5976, this indicates that the segregation is not serious and the modified asphalt has acceptable storage stability. The equation for calculating the softening-point difference is:(2)$$\Delta \mathrm{SP}=\left|{\mathrm{SP}}_{t}-{\mathrm{SP}}_{b}\right|$$
where ΔSP refers to the softening-point difference between the top and bottom of separating tube, ${\mathrm{SP}}_{t}$ and ${\mathrm{SP}}_{b}$ are the softening points of the top and bottom of the separating tube, respectively. Figure 15 illustrates the softening-point difference of BRA-MA. The softening-point difference of BRA-MA with more than 20% BRA-5 was over 2.2 °C, which indicates their severe segregation. Softening-point difference of BRA-MA with BRA-1~4 was lower than 2.2 °C, suggesting that their segregation is slight. However, it was found that the segregation of BRA-MA with BRA-2, -3, and -4 was also severe by manual inspection, which is contrary to the evaluation standard of the softening-point difference test. Additionally, even if the softening-point difference of two kinds of asphalt is equal, their segregation degrees are thought to be different since their original softening points are different. Therefore, the softening-point difference method may not quantitatively indicate the degree of segregation of BRA-MA.

The variance analysis results show that the F value of the softening-point difference by BRA-content dependency was lower than F crit, proving that BRA content did not show a statistical effect on the softening-point difference. In addition, F-value by particle-size dependency was higher than F crit and the *p*-value was less than 0.05, which implies that particle size can significantly affect softening-point difference. Therefore, the softening-point difference test failed to reveal how the BRA content influenced the storage stability, nor can it reveal the quantitative segregation of BRA-MA.

#### 3.2.2. IS Based on the Viscosity Difference

IS based on the viscosity difference was used as an indicator for the purpose of developing the susceptibility of detecting the segregation of BRA-MA. As shown in Figure 5, this evaluation method is also based on the separating-tube method. The viscosity of the top and bottom part of separating-tube should be tested at 135 °C. Higher viscosity signifies the corresponding higher BRA content, according to Figure 13. Additionally, IS was used to illustrate the segregation of BRA-MA. The calculation equation is shown in Equations (3) and (4).
(3)$$\Delta \eta =\left|\eta_{t}-\eta_{b}\right|$$
(4)$$\mathrm{IS}=\Delta \eta /\eta_{O}$$
where Δη refers to the viscosity difference, $\eta_{t}$ and $\eta_{b}$ are the viscosities of the top and bottom of the separating tube, $\eta_{O}$ is the original viscosity of BRA-MA before segregation, and IS is the index of segregation based on the viscosity difference. A higher IS value means a more serious segregation of BRA. Not only the viscosity difference, but also the original viscosity of the modified asphalt had been considered in the calculation of IS. The segregation of asphalt with different original viscosities can be semi-quantitatively compared through the IS method.

Figure 16 presents the IS of BRA-MA after the segregation process, which kept BRA-MA at 165 °C for 48 h. The value of IS showed an almost rising trend, along with a the BRA content increases. Moreover, the IS value illustrated an increasing tendency with the increment in BRA particle size. The F-value is the statistic of the F-test, which is used to indicate the statistical significance of the two methods. The variance analysis results indicate that both the F-values of IS by BRA content and particle-size dependency are higher than the corresponding F crit. The *p*-values proved that the discrepancy of IS by BRA content and particle-size dependency were significant. Consequently, IS based on the viscosity difference had more static significance in the evaluating influence of BRA content and particle size on segregation compared with the softening difference evaluation. IS was then determined as the method to indicate the storage stability of BRA-MA in the following study.

### 3.3. Static Storage Stability in the Production Plant

Storage temperature and time are the key factors affecting the static storage stability of samples. Therefore, how BRA content and particle size affect static storage stability at different temperatures and times was studied. The storage temperatures were 145 °C, 155 °C, and 165 °C, while the storage times were 24, 48, and 72 h. The aging of BRA-MA at high temperatures was negligible in a soaking tank, which cut off the outside air.

Figure 17, Figure 18 and Figure 19 present the IS results of BRA-MA after the segregation process by storage temperature and time dependency, respectively. The IS values illustrated an obviously rising trend as the BRA content increased, which also showed an increasing tendency along with particle-size increase from BRA-1 to BRA-5. On the other hand, with the extension of storage time, the IS value also showed an increasing tendency It was evident that heightening temperatures would result in higher IS values by comparison of Figure 17, Figure 18 and Figure 19. Therefore, reducing the temperature, particle size, storage time, and BRA content would help to lower the segregation of BRA-MA in static storage in a static container.

The correlation between the specific surface area of BRA and IS was analyzed. Figure 20 presents the IS results by temperature, specific surface area, and BRA content at 145 °C, 155 °C, and 165 °C, respectively, for 24 h’ static storage, for instance. The IS value presented a decline as the increment in the specific surface area of BRA, since its particle size was negatively correlated to the specific surface area. Power function curves were used to fit the data point. The result of the fitting equation and R^2^ of BRA-MA after the segregation process for 24 h of static storage is shown in Table 3. Corresponding complex correlation coefficients (R^2^) were higher than 0.89, suggesting that fitting between the IS value and specific surface area was rational. It revealed that the functional relationship of IS and specific surface area can be fit as Equation (5):(5)$$\mathrm{IS}=s_{w_{i}}x^{-a}$$
where $s_{w_{i}}$ refers to the segregation coefficients, which is positively correlated to the segregation of BRA-MA; x is the variable (specific surface area); and a is the power. This functional relationship of the IS and specific surface area can be used to indicate how particle size that is correlated to the specific surface area determines the segregation of BRA-MA. Table 4 listed the Power function curve-fitting equation and R^2^ of BRA-MA. It is obvious that the R^2^ is mostly over 0.9, which suggested that the fitting is adequate.

### 3.4. Storage Stability during Transportation

Transportation is a non-negligible process for storage stability and the aging of modified asphalt [32,33]. A simulated experiment was designed to investigate the storage stability variation of BRA-MA during its transportation. BRA-MA was pumped from a static container into a vehicle container that was in a delivery vehicle in a production plant. The original temperature of BRA-MA in the container was found to be around 145 °C due to heat loss in the pumping process. Then, the vehicle should transport BRA-MA to the corresponding construction site (as Figure 3 illustrated). The transportation time is usually less than 12 h, according to the on-the-spot investigation. The heat loss of BRA-MA during the transportation process can be alleviated by the insulation layer and heating equipment of the containers on the vehicles. The temperature decline in BRA-MA of 12 h during the transportation process was 20 °C, which means its temperature decreasing rate is averagely of 5 °C per 3 h. This experiment consequently simulated the segregation of BRA-MA in consideration of its cooling process during transportation.

Figure 21 explains the following stages of this simulated experiment. The original temperature of BRA-MA in the separating tubes was 145 °C. Then, BRA-MA should be placed into a temperature-controlling box, and the initial temperature is 145 °C when the time is marked as 0 h. The temperature of the box should be switched to 140 °C when the storage time is 1.5 h. Then, the temperature of the BRA-MA should be tested for the first time at 3 h to confirm that its temperature has already decreased to 140 °C. Meanwhile, the IS value at 140 °C should be tested according to the viscosity difference test conducted as the temperature test ①. As Figure 21 illustrates, the temperature and IS value of BRA-MA should be tested when its temperature gradually reduces to 135, 130, and 125 °C, as the temperature test ②, ③ and ④, respectively. The temperature of BRA-MA finally decreased to 125 °C after 12 h; the cooling-process simulation of the transportation was terminated.

The IS values of BRA-MA with BRA-1~4 during transportation are provided in Figure 22, while the IS value of BRA-5-based BRA-MA is shown in Figure 23. The abscissa is the temperature during transportation, and its corresponding transportation times are 3, 6, 9, and 12 h. The results show that the IS values of BRA-MA with BRA-1~4 are below 0.2, suggesting that their segregation is very slight. The IS value also showed an increasing trend as the particle size rises. However, BRA-5-based BRA-MA illustrated a high IS value during transportation, proving that its segregation was serious. In addition, the IS values of BRA-MA with BRA-5 presented an increasing tendency in the cooling process. However, IS change of BRA-1~4-based BRA-MA in the cooling process was not explicit, especially when BRA content was below 30%. Therefore, BRA-MA with BRA-1~4 showed low segregation and acceptable storage stability compared to BRA-5-based BRA-MA, although their IS values changed unreasonably. BRA-MA with BRA-1~4 whose d(0.5) particle sizes were lower than 13.6 μm showed low segregation. However, the segregation of BRA-MA with BRA whose d(0.5) particle size was over 106 μm was severe.

## 4. Conclusions

This study introduced BRA of different particle sizes and contents to prepare BRA-MA in order to determine how the factors affect their physical properties and storage stability. Storage-stability evaluation methods were also discussed. Based on the results, the following conclusions can be drawn.

(1)Softening point was positively correlated to BRA content, while the particle size of BRA showed a negative correlation. Penetration of BRA-MA increased as the increment in particle size, while BRA content negatively affected penetration. Both BRA content and particle size negatively determines the ductility of BRA-MA. A larger particle size of BRA resulted in lower viscosity, but a higher BRA content increased the viscosity of BRA-MA. The temperature susceptibility of BRA-MA decreased with the increase in BRA content, but the smaller particle size led to lower temperature susceptibility.(2)Separating-tube method was used to simulate the segregation of BRA-MA in Lab. How BRA-content influence on storage stability could not be significantly reflected by the softening-point difference test, and the segregation degree was not quantitively revealed. The IS value based on the viscosity difference test had a higher statistical significance when evaluating the influence of BRA content and particle size on segregation. The segregation of asphalt with different original viscosity could be quantitatively compared through the IS value based on the viscosity difference test.(3)Storage stabilities of static and transportation corresponded to storage in a production plant and during the vehicle transportation process. The segregation of BRA-MA illustrated a rising trend as the BRA coffintent and particle size increased. Both storage temperature and time were positively correlated to the segregation of BRA-MA. It was proved that the relationship between the specific surface area and segregation were power functional. This relationship can be used to understand how particle size, which is correlated to specific surface area, determines the segregation of BRA-MA. Based on the simulated experiment of transportation segregation, BRA-MA with BRA-1~4 whose d(0.5) particle sizes ere lower than 13.6 μm, showed low segregation. However, the segregation of BRA-MA with BRA whose d(0.5) particle size was over 106 μm was severe.

## Figures and Tables

**Figure 1 materials-15-03592-f001:**
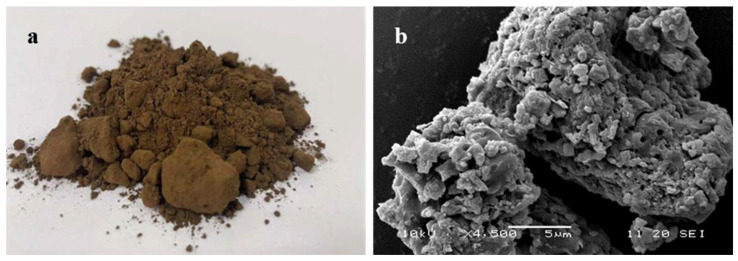
BRA powder (**a**) and its SEM image (**b**).

**Figure 2 materials-15-03592-f002:**
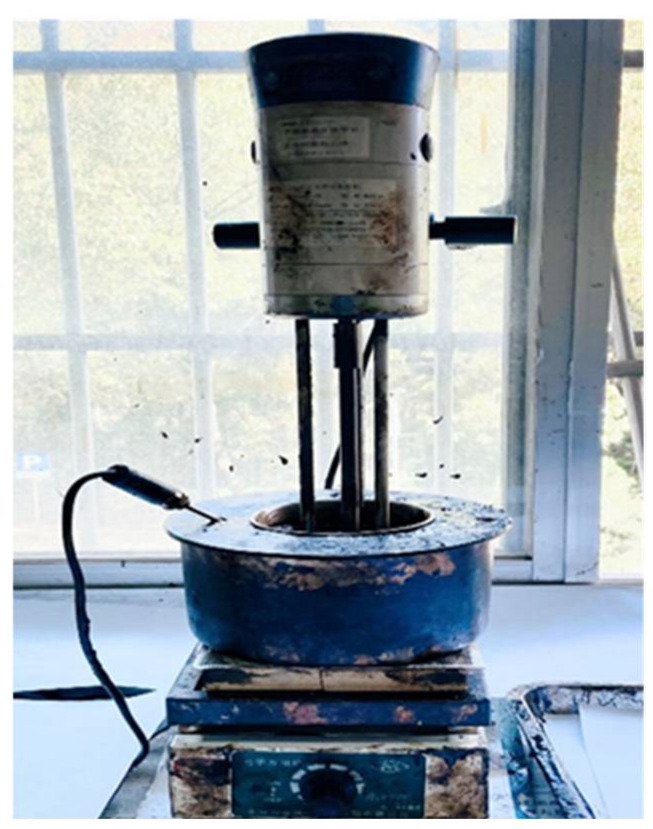
High-speed shear agitator for BRA-MA preparation.

**Figure 3 materials-15-03592-f003:**
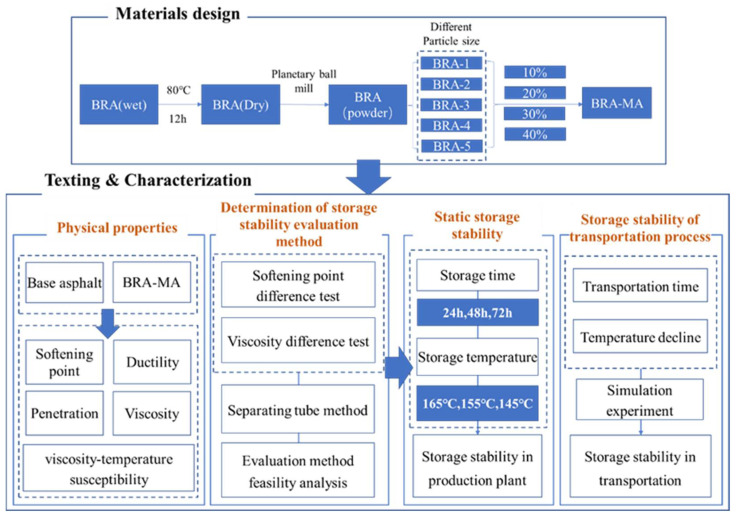
Outline of this study.

**Figure 4 materials-15-03592-f004:**
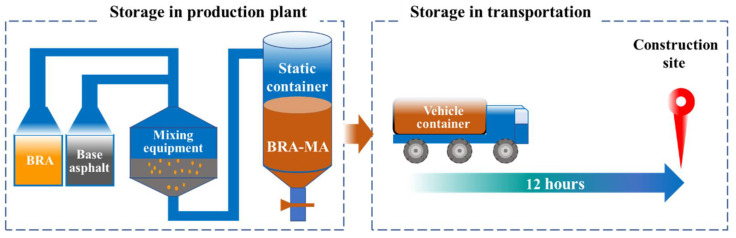
Storage process of BRA-MA in a production plant and during transportation.

**Figure 5 materials-15-03592-f005:**
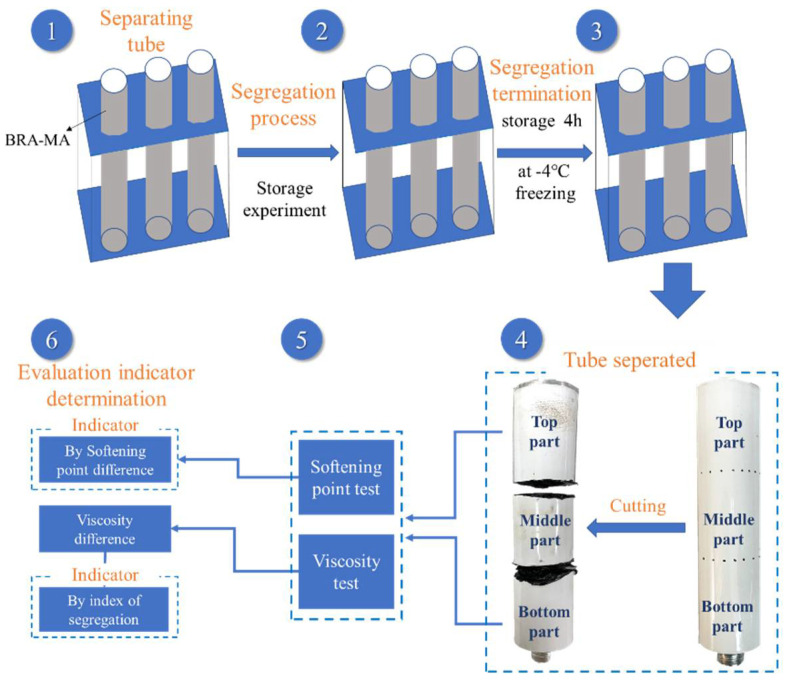
Separating-tube method of index of storage stability.

**Figure 6 materials-15-03592-f006:**
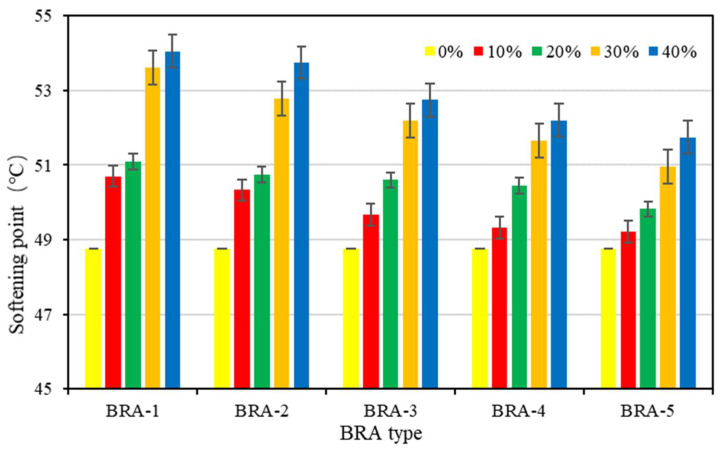
Softening point of base asphalt and BRA-MA.

**Figure 7 materials-15-03592-f007:**
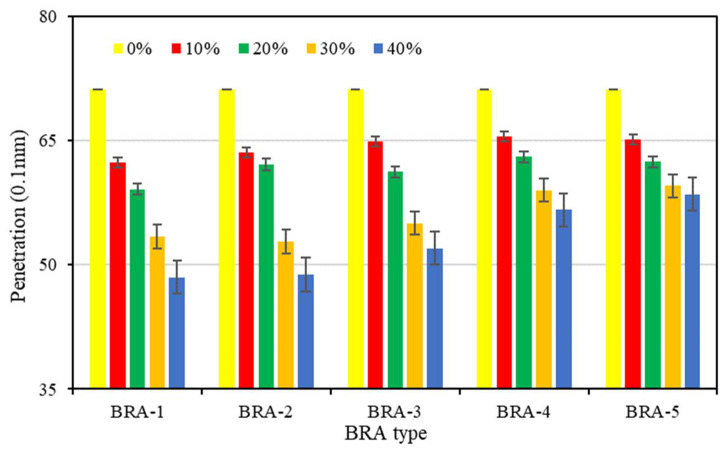
Penetration of base asphalt and BRA-MA.

**Figure 8 materials-15-03592-f008:**
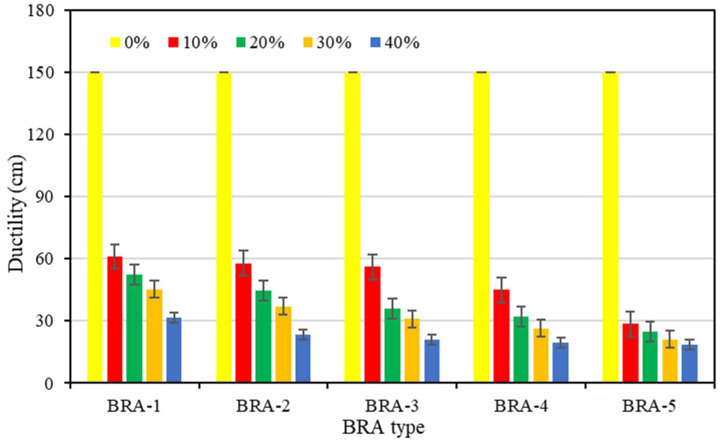
Ductility of base asphalt and BRA-MA.

**Figure 9 materials-15-03592-f009:**
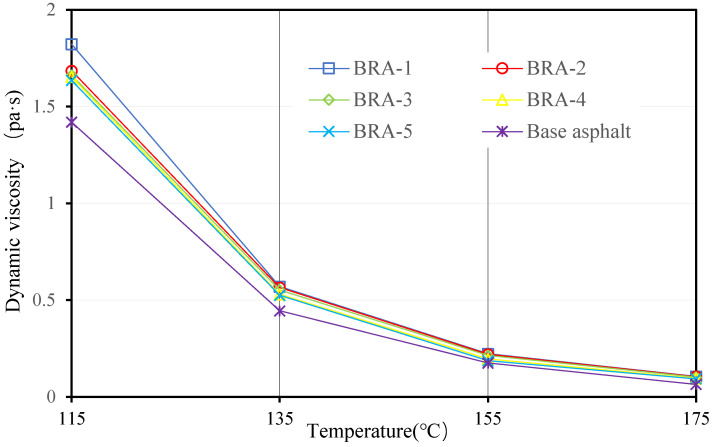
Viscosity of BRA-MA with 10% BRA.

**Figure 10 materials-15-03592-f010:**
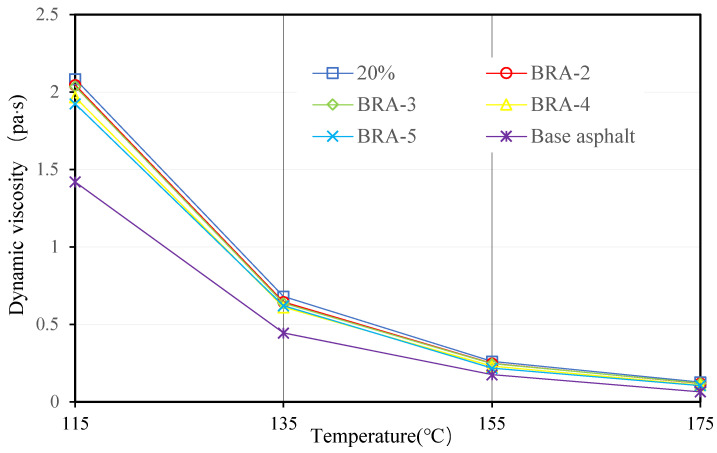
Viscosity of BRA-MA with 20% BRA.

**Figure 11 materials-15-03592-f011:**
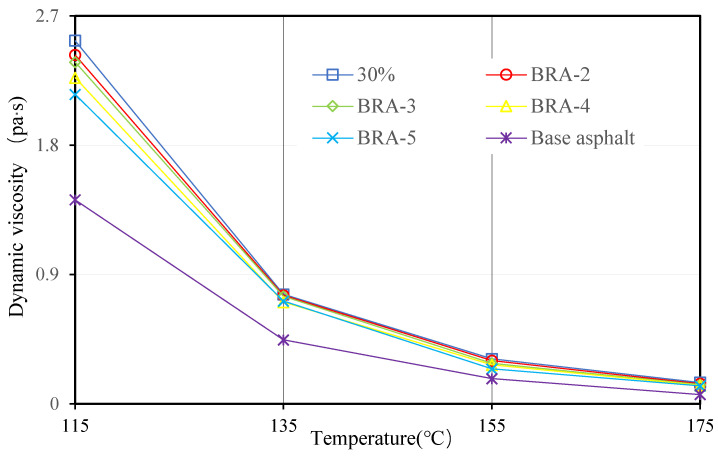
Viscosity of BRA-MA with 30% BRA.

**Figure 12 materials-15-03592-f012:**
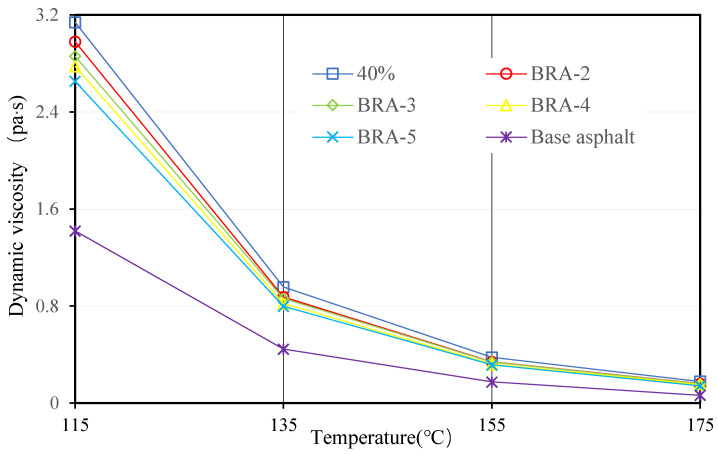
Viscosity of BRA-MA with 40% BRA.

**Figure 13 materials-15-03592-f013:**
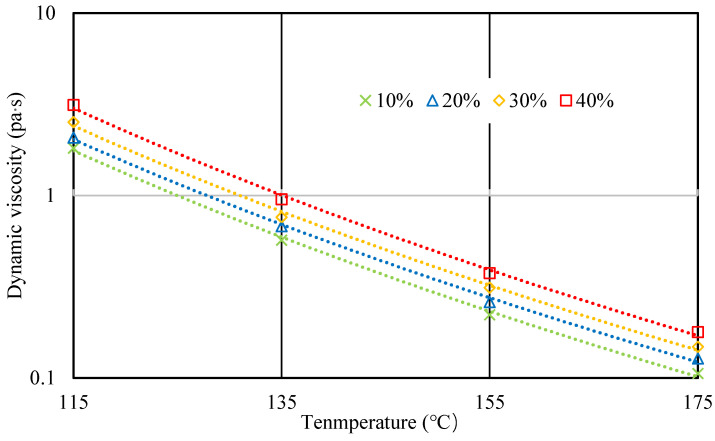
Viscosity of BRA-MA with 40% BRA with logarithmic axis.

**Figure 14 materials-15-03592-f014:**
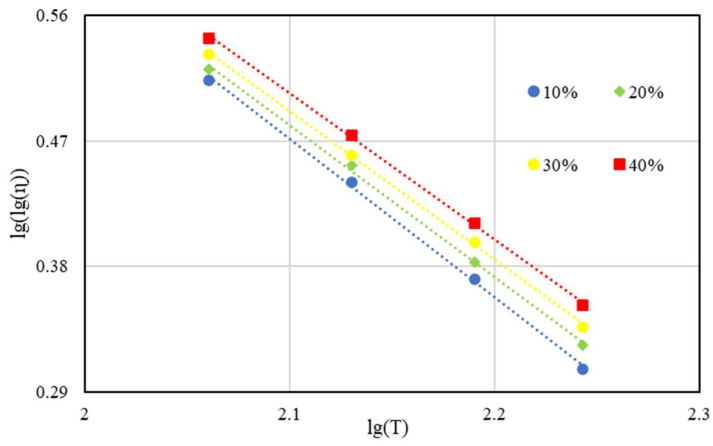
Viscosity–temperature curve of BRA-MA with BRA-1 for VTS.

**Figure 15 materials-15-03592-f015:**
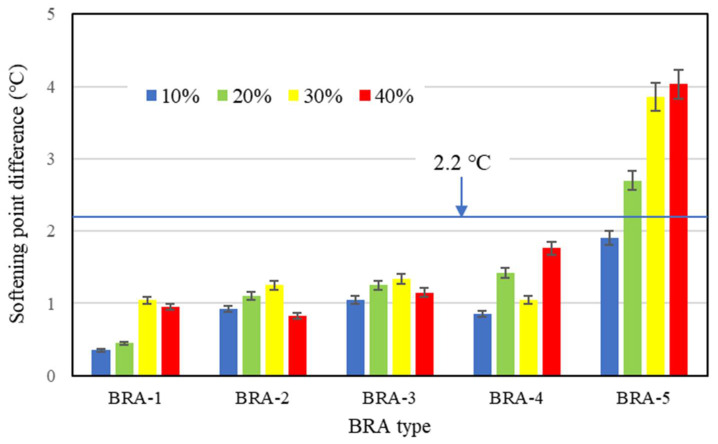
Softening-point difference of BRA-MA.

**Figure 16 materials-15-03592-f016:**
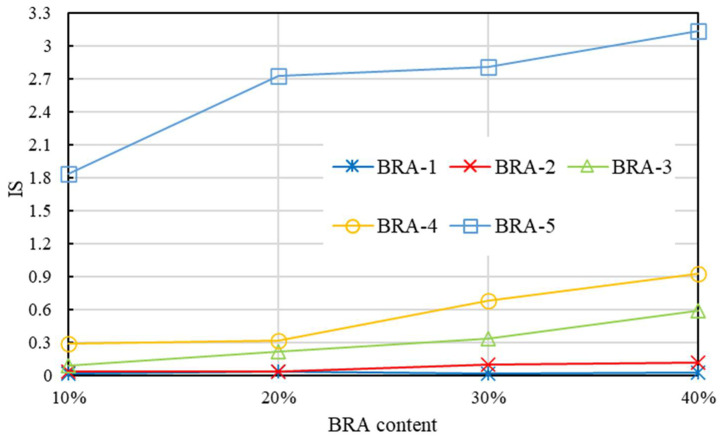
IS of BRA-MA after the segregation process.

**Figure 17 materials-15-03592-f017:**
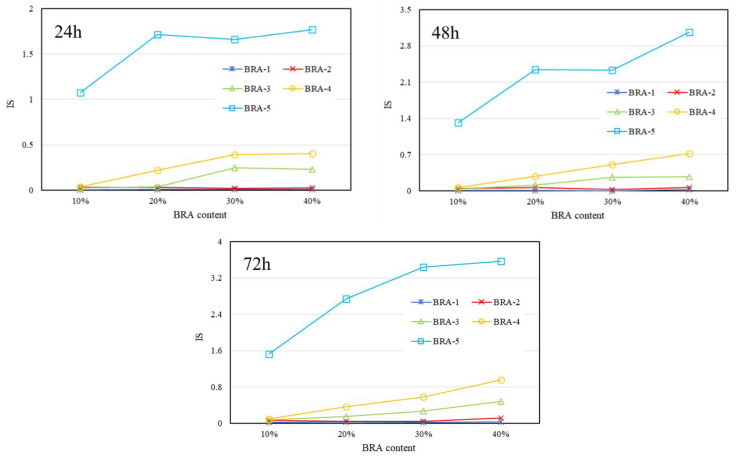
IS of BRA-MA after the segregation process at 145 °C.

**Figure 18 materials-15-03592-f018:**
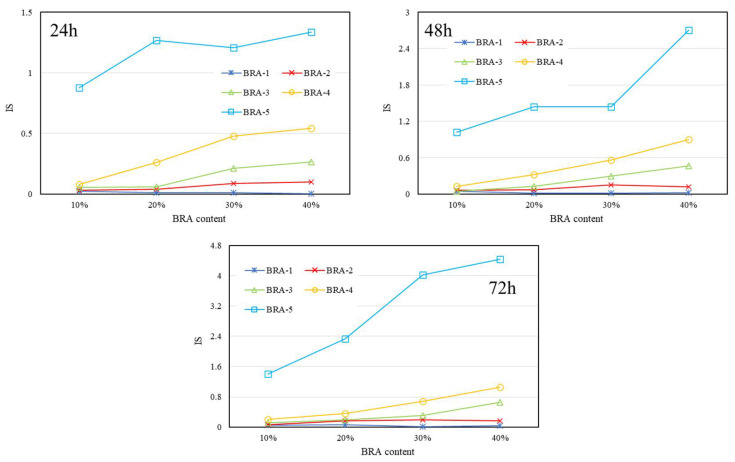
IS of BRA-MA after the segregation process at 155 °C.

**Figure 19 materials-15-03592-f019:**
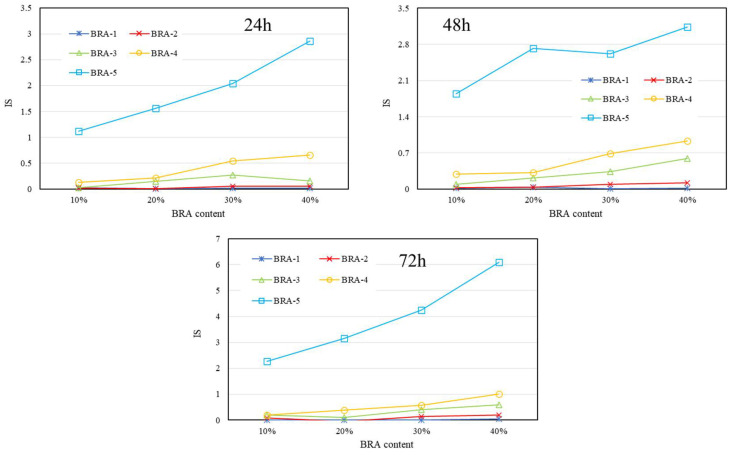
IS of BRA-MA after the segregation process at 165 °C.

**Figure 20 materials-15-03592-f020:**
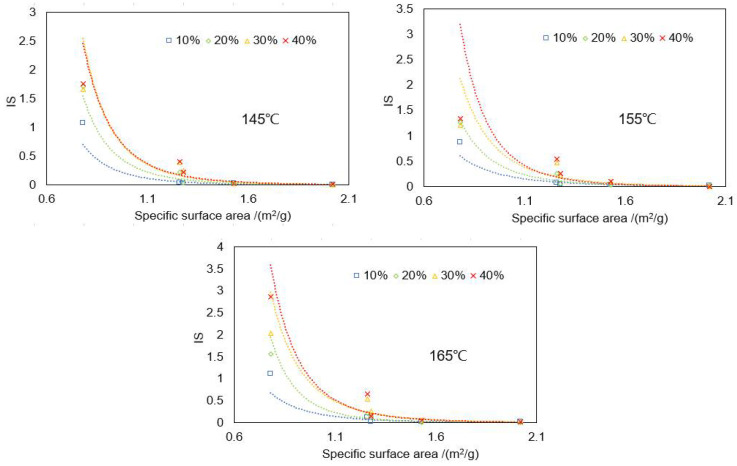
IS of BRA-MA after the segregation process for 24 h static storage at 145 °C, 155 °C, and 165 °C.

**Figure 21 materials-15-03592-f021:**
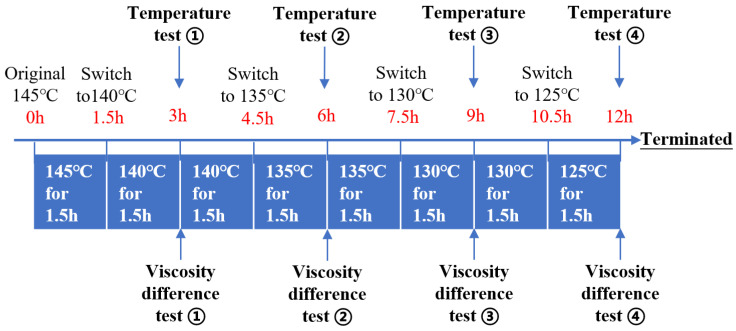
Outline of simulated experiment for the storage stability during transportation.

**Figure 22 materials-15-03592-f022:**
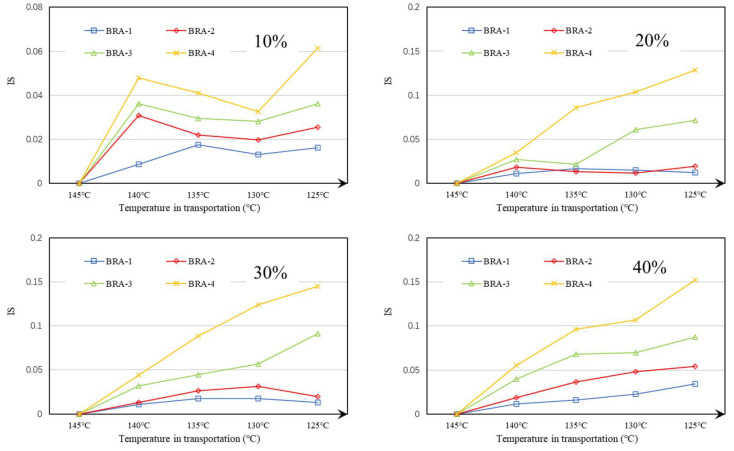
IS of BRA-MA with BRA-1~4 during transportation.

**Figure 23 materials-15-03592-f023:**
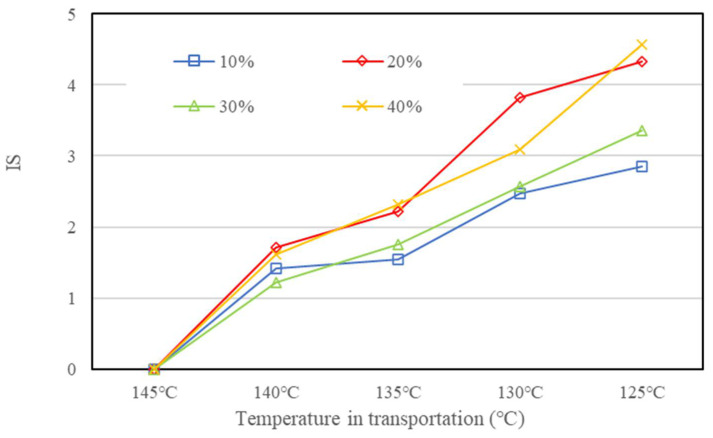
IS of BRA-MA with BRA-5 during transportation.

**Table 1 materials-15-03592-t001:** Properties of base asphalt.

Properties	Penetration (25 °C, 100 g, 5 s)	Softening Point	Ductility (5 cm/min, 15 °C)	Brookfield Viscosity (10 r/min, 135 °C)
Unit	0.1 mm	(°C)	cm	Pa·s
Base asphalt	71.2	48.8	≥100	0.142
Standard limits	60~80	≥46.0	≥100	-

**Table 2 materials-15-03592-t002:** Particle-size testing results of BRA.

BRA Number	d(0.1) (μm)	d(0.5) (μm)	d(0.9) (μm)	Specific Surface Area (m^2^/g)
BRA-1	1.20	6.26	21.07	2.02
BRA-2	1.84	9.55	23.87	1.53
BRA-3	2.39	12.58	32.27	1.28
BRA-4	2.26	13.60	37.64	1.26
BRA-5	75.29	106.22	150.55	0.78

**Table 3 materials-15-03592-t003:** Absolute value of BRA-MA VTS values.

	BRA-1	BRA-2	BRA-3	BRA-4	BRA-5
0%	1.297	1.297	1.297	1.297	1.297
10%	1.135	1.117	1.128	1.164	1.176
20%	1.087	1.112	1.114	1.147	1.166
30%	1.066	1.083	1.104	1.107	1.120
40%	1.048	1.074	1.075	1.081	1.097

**Table 4 materials-15-03592-t004:** Power function curve-fitting equation and R^2^ of BRA-MA.

Temperature (°C)	BRA Content (%)	Power Function Curve-Fitting Equation	R^2^
145	10	y = 0.1929x^− 5.216^	0.9968
	20	y = 0.3852x^− 5.58^	0.9930
	30	y = 0.6138x^− 5.727^	0.9672
	40	y = 0.6268x^− 5.505^	0.9737
155	10	y = 0.2284x^− 3.913^	0.9935
	20	y = 0.3813x^− 4.937^	0.9816
	30	y = 0.6213x^− 4.967^	0.9104
	40	y = 0.7397x^− 5.889^	0.8968
165	10	y = 0.2222x^− 4.498^	0.9928
	20	y = 0.4174x^− 6.197^	0.9915
	30	y = 0.8069x^− 5.256^	0.9641
	40	y = 0.8867x^− 5.629^	0.9713

## Data Availability

Not applicable.

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
