# Peer review of "Physical Properties and Storage Stability of Buton Rock Asphalt Modified Asphalt"

_materials, 2022, doi:10.3390/ma15103592_

Round 1

Reviewer 1 Report

The topic is good but improvement required. 

  1. Many error in writing and citing the reference/table/figure.
  2. Many grammar and spelling error. English proofread is a required.
  3. Lack of discussion, author only reporting the results without discussing the reason and comparing with previous study.

Overall, major revision is required. 

Author Response

The topic is good but improvement required. 

Response: Thanks for your comments. Revision had been conducted according to your comments. The writing had been improved by seeking help from a native English speaker.

  1. Many error in writing and citing the reference/table/figure.

Response: Thanks for your comments. The errors had been corrected according to your marks and self-examination.

  1. Many grammar and spelling error. English proofread is a required.

Response: Thanks for your comments. The grammar and spelling error had been checked and corrected.

  1. Lack of discussion, author only reporting the results without discussing the reason and comparing with previous study.

Response: Thanks for your comments. We deeply agree that the discussion should be presented in addition to revealing objective laws. The analysis had been added in the manuscript as possible. However, there are few studies about the static and transportation storage stability in previous study, hence it is hard to analyze it by comparing with them. Additionally, the widely accepted standard of BRA is still not reported. More discussion will be presented in the further work after in-depth study of the BRA-MA.

Reviewer 2 Report

Manuscript name: Physical Properties and Storage Stability of Buton Rock Asphalt Modified Asphalt

Manuscript ID: Materials 2021, 14 x

The manuscript was referred to as Buton Rock Asphalt (BRA) using the different particle sizes and contents

to prepare BRA modified asphalt (BRA-MA). The properties of the BRA-MA were examined by applying the tests during storage and transportation to understand the segregation process. Conventional tests such as viscosity, softening point, ductility, and penetration were performed in the manuscript and reinforced with the function equation. The test results were evaluated by considering the material particle size, material ratios, temperature, and time changes as well. According to the results, it was seen that not only the particle size and content but also storage temperature and time were positively correlated to the segregation of BRA-MA.

In the evaluation of the aforementioned study, the following query and suggestions should be considered.

1-         The manuscript title should be changed to a better explanation.

2-         The author should share a few pictures of the experimental work.

3-         A new study has been done on this subject in 2021-2022, this can be referenced.

4-         The BRA should be briefly explained in the abstract.

5-         Repeated sentences should be avoided.

6-         Materials (2.1) part must be rearranged due to incorrect expression. (Error! Reference source not found).

7-         It should be explained in the abstract (d(0,5)?)

8-         It should be shown  BRA powder bonds in SEM (Figure 1).

9-         Standard types and standard limits must be given in Table 1.

10-       The manuscript should be grammatically revised and spelling mistakes should be corrected (page 4; in in, page 16; xis, page 19; BRAMA, and so on).

11-       Figure 4 should be revised to be more readable.

12-       Experimental studies should be given together with standards and evaluated according to the specification limits (Result and Discussion)

13-       Figure 5 should be revised to the standard limit and threshold values should be indicated.

14-       Penetration (3.1.2) part must be rearranged due to incorrect expression. (Error! Reference source not found).

15-       Figure 6 should be revised to the standard limit and threshold values should be indicated.

16-       which type of BRA is used on page 7. It should be stated ductility part (3.1.3 ductility)

17-       It should be arranged on page 8 due to there is no standard given in this part (title of 3.1.4.).

18-       Figure 7 should be rearranged with the standard limit.

19-       The first paragraph of title 3.2 should be transferred to Figure 2.

20-       Abbreviations should be specified in the formula (Page 13).

21-       It should be reduced the Y-axis clearance in Figure 15 because curves are not better understood.

22-       It should be explained what is meant by the F value (Page 13).

23-       Figure 19 should be rearranged to better understand

24-       Figure 22 must be checked in the manuscript (20%, 40%)

Consequently; the topic is interesting and a huge amount of work including analyses have made in the manuscript. The validation and verification of the model are clear. For this reason, in the reviewer's opinion; the manuscript needs to minor revision due to the comments reported above.

Author Response

In the evaluation of the a forementioned study, the following query and suggestions should be considered.

Response: Thanks for your comments. Revisions had been conducted in the manuscript according to your comments.

1-         The manuscript title should be changed to a better explanation.

Response: Thanks for your comments. This manuscript focused on how the BRA particle size and content affect the physical performance and storage stability. Therefore, the title “Physical Properties and Storage Stability of Buton Rock Asphalt Modified Asphalt” is thought to be able to summarize the work of this manuscript.

2-         The author should share a few pictures of the experimental work.

Response: Thanks for your comments. Picture of high-speed shear agitator for BRA-MA preparation had been added in the text.

3-         A new study has been done on this subject in 2021-2022, this can be referenced.

Response: Thanks for your comments. Several study which containing relevant contents in 2021-2022 had been referenced.

4-         The BRA should be briefly explained in the abstract.

Response: Thanks for your comments. The explanation of BRA had been briefly explained in abstract.

5-         Repeated sentences should be avoided.

Response: Thanks for your comments. The repeated sentence had been checked and corrected.

6-         Materials (2.1) part must be rearranged due to incorrect expression. (Error! Reference source not found).

Response: Thanks for your comments. This is an error of citation, the errors had been corrected.

7-         It should be explained in the abstract (d(0,5)?)

Response: Thanks for your comments. The sentence in abstract “The BRA-MA with BRA whose d(0.5) are lower than 13.6 μm showed low segregation in transportation.” Had been replaced by “The BRA-MA with BRA whose 50% particle sizes are lower than 13.6 μm showed low segregation in transportation.” to make it better to understand.

8-         It should be shown BRA powder bonds in SEM (Figure 1).

Response: Thanks for your comments. It is true that the agglomeration of BRA powder is found in a part of Figure 1. But the bond of BRA power was not obvious since the crushed BRA powder was dried and ground before SEM characterization. Consequently, the bonding of the powder cannot be shown in SEM image.

9-         Standard types and standard limits must be given in Table 1.

Response: Thanks for your comments. The standard and standard limits had been added in the text and table as possible.

10-       The manuscript should be grammatically revised and spelling mistakes should be corrected (page 4; in in, page 16; xis, page 19; BRAMA, and so on).

Response: Thanks for your comments. The grammar and spelling error had been checked and corrected.

11-       Figure 4 should be revised to be more readable.

Response: Thanks for your comments.  The expression of Figure 4 had been improved to be better readable.

12-       Experimental studies should be given together with standards and evaluated according to the specification limits (Result and Discussion)

Response: Thanks for your comments. Some standard limits of BRA-MA properties are hard to define due to the lack of widely accepted specification for rock asphalt modified asphalt. Corresponding standard and specification limits had been added in the test of Result and Discussion as possible.

13-       Figure 5 should be revised to the standard limit and threshold values should be indicated.

Response: Thanks for your comments. The standard had been added.

14-       Penetration (3.1.2) part must be rearranged due to incorrect expression. (Error! Reference source not found).

Response: Thanks for your comments. The error had been checked and corrected.

15-       Figure 6 should be revised to the standard limit and threshold values should be indicated.

Response: Thanks for your comments. Corresponding standard had been given as possible. However, it is very hard to give a standard limit for the Figure 6 since there is no standard limits for the penetration of BRA-MA yet.

16-       which type of BRA is used on page 7. It should be stated ductility part (3.1.3 ductility)

Response: Thanks for your comments. The BRA throughout this manuscript have 5 different particle size (BRA-1,BRA-2, BRA-3, BRA-4 and BRA-5) and 4 content (10%, 20%, 30% and 40%). Corresponding explanation had been listed in “2.2. Preparation of BRA-MA” and Table 2. Therefore, special instructions were not added in the result and discussion.

17-       It should be arranged on page 8 due to there is no standard given in this part (title of 3.1.4.).

Response: Thanks for your comments. The standard had been given in this paragraph.

18-       Figure 7 should be rearranged with the standard limit.

Response: Thanks for your comments. Since there is no widely accepted standard limits for the ductility of BRA-MA, the specification limit can not be indicated in this Figure. We will thereby devote ourselves to determining more scientific and appropriate specification for BRA-MA in future research.

19-       The first paragraph of title 3.2 should be transferred to Figure 2.

Response: Thanks for your comments. This recommendation is very helpful. The first paragraph of title 3.2 had been transferred to Figure 3(which is Figure 2 in previous manuscript).

20-       Abbreviations should be specified in the formula (Page 13).

Response: Thanks for your comments. The Abbreviations had been specified.

21-       It should be reduced the Y-axis clearance in Figure 15 because curves are not better understood.

Response: Thanks for your comments. The Y-axis clearance in Figure 15 had been reduced.

22-       It should be explained what is meant by the F value (Page 13).

Response: Thanks for your comments. F value is the statistic of F test, which is used to indicate the statistical significance of the two methods. Corresponding explanation had been added in the paragraph.

23-       Figure 19 should be rearranged to better understand

Response: Thanks for your comments. The Figure had been rearranged.

24-       Figure 22 must be checked in the manuscript (20%, 40%)

Response: Thanks for your comments. We had checked these data and found it is correct.

Consequently; the topic is interesting and a huge amount of work including analyses have made in the manuscript. The validation and verification of the model are clear. For this reason, in the reviewer's opinion; the manuscript needs to minor revision due to the comments reported above.

Response: Thanks for your comments. We had revised this manuscript in accordance with your comments.

Reviewer 3 Report

In this study, BRA of different particle sizes and contents to prepare the BRA-MA in order to find how the factors affect their physical properties and storage stability were presented. In my opinion, this paper can consider to publish on materials journal but Authors has to revise this paper based on my comments:

  1. Should improve the abstract: introduction, problem statement, aim, method, results, and conclusion
  2. For Page 3 has “Error! Reference source not found “
  3. For Page 7 has “Error! Reference source not found “
  4. For Page 8 has “Error! Reference source not found
  5. For Page 11 has “Error! Reference source not found
  6. Any properties of Buton Rock Asphalt Modified Asphalt? how proper is the mixing process?
  7. Testing program is not properly discussed and written
  8. You need to conduct DSR test?
  9. Lack of results discussion anything to do with the Buton Rock addition? how does it affects the performance?

Author Response

In this study, BRA of different particle sizes and contents to prepare the BRA-MA in order to find how the factors affect their physical properties and storage stability were presented. In my opinion, this paper can consider to publish on materials journal but Authors has to revise this paper based on my comments:

  1. Should improve the abstract: introduction, problem statement, aim, method, results, and conclusion

Response: Thanks for your comments. We had checked the abstract and improved it.

  1. For Page 3 has “Error! Reference source not found “

Response: Thanks for your comments. The error had been corrected.

  1. For Page 7 has “Error! Reference source not found “

Response: Thanks for your comments. The error had been corrected.

  1. For Page 8 has “Error! Reference source not found

Response: Thanks for your comments. The error had been corrected.

  1. For Page 11 has “Error! Reference source not found

Response: Thanks for your comments. The error had been corrected.

  1. Any properties of Buton Rock Asphalt Modified Asphalt? how proper is the mixing process?

Response: Thanks for your comments. The properties of BRA is hard to define since there is no formal specification for characterizing the BRA. The result showed that the mass percentage of asphalt in BRA was 27.8%, while the minerals account for 71.2%. The mixing is found to be adequate since the high-speed shear agitator introduced in this study has good mixing function.

  1. Testing program is not properly discussed and written

Response: Thanks for your comments. We agree that the testing program especially the testing method of storage stability requires revision. Therefore, the section of 2.3. Experimental methods had been improved.

  1. You need to conduct DSR test?

Response: Thanks for your comments. DSR test is not required according to the topic of this manuscript. Rheological properties of the BRA-MA will tested in the further work.

  1. Lack of results discussion anything to do with the Buton Rock addition? how does it affects the performance?

Response: Thanks for your comments. We had added the discussion on how the BRA affect the performance of BRA-MA as possible. However, there are few studies about the static and transportation storage stability in previous study, hence it is hard to analyze it by comparing with them. Additionally, the widely accepted standard of BRA is still not reported. More discussion will be presented in the further work after in-depth study of the BRA-MA.

Round 2

Reviewer 3 Report

Good job!